# Renewable Energy and Sustainability from the Supply Side: A Critical Review and Analysis

**Susana Garrido \*, Tiago Sequeira and Marcelo Santos**

Faculty of Economics, University of Coimbra (CEBER), Av. Dias da Silva, 165, 3004-512 Coimbra, Portugal; tiago.n.sequeira@fe.uc.pt (T.S.); marcelosantos@uc.pt (M.S.)
\* Correspondence: garrido.susana@gmail.com

**Abstract:** This article provides a critical review of the literature on the relationship between renewable energies and sustainability considering the three dimensions of sustainability: economic, social, and environmental. First, a bibliometric tool is used and then a more in-depth analysis of selected literature is performed, focusing on the type of renewable energy analyzed and the level of development of countries, the dimension of sustainability focused on and the country's development level, and the type of renewable energies focused on and the dimension of sustainability analyzed. It represents a milestone in the topic giving insights on the state of the art of the research on this research area, enhancing empirical evidence on the kind of relationships and developing a discussion on how closely aligned the political and institutional discourses are with the research concerns. We conclude that, while studies on lower-income countries focus on lower-rung energies, studies on higher-income countries focus on the study of more diversified sources. Moreover, wind–solar energy is the most reported in the articles concerned with environmental sustainability. Our main recommendation is to further investigate the implementation of modern renewable energies in developing countries, to help those countries to climb the energy ladder toward cleaner energy supply.

**Keywords:** renewable energies; sustainability; review; energy resources

## 1. Introduction

The concepts of sustainable development and sustainability are interchangeable, and usually include three dimensions: economic, social, and environmental [1]. The concept of sustainable development was first provided in the report of the Bruntland Commission, where it was defined as "meeting the needs of the present generation without compromising the ability of future generations to meet their own needs" [2], meaning that economic growth, social inclusion, and environmental protection are the three main pillars of sustainable development [3]. The initial concept of sustainability was associated with environmental concerns, focused specifically on the preservation of resources. This has now become a milestone for the entire business community. For example, Herbohn et al. [4] warned of the risk of extinction of iconic species or loss of entire ecosystems and water resource threats. Among the most widely acknowledged definitions of sustainability is the so-called triple bottom line (TBL), in which economic, social, and environmental responsibility are emphasized [5].

One of the most effective ways to achieve sustainability targets is to reduce energy consumption, along with its many adverse consequences. There are two fundamental ways to do so: employ energy saving measures or use renewable energy (RE) to generate power. As these energy-related measures contribute so heavily to sustainability, investors find RE technologies very attractive. Discovering and implementing new technologies are important steps in the provision of cheap, reliable, ecologically sound, and accessible energy around the world [6].

Painuly [7] identified a set of factors that should be considered while assessing the development of RE, mainly the techno-economic and economic factors. Moreover, numerous studies indicate different factors that exert an impact on RE development. For example, Aguirre and Ibikunle [8] included socio-economic factors and countries' income (Gross Domestic Product (GDP) growth or GDP per capita). In the Renewables 2018 Global Status Report [9], the statistics associated with RE power production capacities in 2017 show that the biggest economies now produce most of the RE (BRICs (429 Gigawatts), EU-28 (320 Gigawatts), China (334 Gigawatts), USA (161 Gigawatts), and Germany (106 Gigawatts)) since they are more inclined to invest in some form of RE [10]. These last authors pointed out that the domestic availability of conventional fossil fuel resources in most developing countries is inadequate, which forces those countries to import energy or the fuel to produce it. Population growth in these countries adds to the demand for imported energy, making the economic situation of the developing countries even more difficult. Apergis and Danuletiu [11] argued that there exists a bidirectional causality between energy consumption and economic growth, pointing out that RE supports economic growth, which in turn encourages the use of more RE. According to these authors, this results in a virtuous cycle, boosting the economy and benefitting society.

Additionally, there is an empirical regularity between development (or per capita income) and the energy ladder (i.e., the gradual replacement of fuel fossil sources of energy by renewable sources). In this line, Ramalho et al. [12] concluded that income growth (associated with democratic countries) encourages the replacement of hydroelectric and oil sources by coal and nuclear and less by natural gas and renewable sources such as biomass, wind, and others. Despite the recognized relationship between renewable energies and the level of development of countries, the research that has been published on this topic is quite narrow in scope.

Bozkurt and Destek [13] analyzed the consumption of renewable and non-renewable energies to explore their impacts on the environment using a case study methodology (24 OECD (Organisation for Economic Co-operation and Development) countries) for the period 1980–2014. In addition, the Alam and Murad [14] investigated the impact of a set of macroeconomic variables such as economic growth on renewable energy use in the short run and long run across 25 OECD countries. Zafar et al. [15], adopting a more general approach, focused on the effects of non-renewable and renewable energy consumption on economic growth of countries.

It is strategic to study the state of the art about the kind of research that has been published that focuses on the relationship between countries' development and the type of RE(s) in a way that makes it possible to identify which RE(s) are studied and implemented in which countries according to their level of development. Thus, introducing RE as an alternative energy would help to keep trade balanced, and securing access to adequate energy supplies is a vital challenge to economic development [16].

The relationship between sustainability and the level of countries' development is an important topic contributing to a better understanding of whether countries with different levels of development also have different sustainability behaviors. According to Juknys et al. [17], in the developed countries, people's well-being does not necessarily follow from economic growth, and social sustainability is sometimes not achieved. From the perspective of the same authors, gradually slowing the rate of economic growth to zero is the natural way to obtain sustainability, especially in countries that greatly exceed their bio-capacity. Thus, the relationship between countries' development and their sustainability has become a fashionable topic.

Another issue arises from the literature review, which reveals a strong belief that the concerns about the different dimensions of sustainability depend on the country on which the study is based. As this topic is so important, it is strategic to know how it has been studied among the scholarly community. Zhang et al. [18] investigated the links between research and development, transport, real income, and transport's $CO_2$ emissions in Organization for Economic Cooperation and Development (OECD) countries from 1990 to 2015. Wang et al. [19] used panel data techniques to analyze the drivers of RE use in a group of 32 countries. Basu and Trica [20] analyzed the sustainability of the circular economy indicators and based on a panel data propose a model for determining the dependency of

the main circular economy factors on EU economic growth. As these works are quite specific in their scope, an important contribution to the research community would be a work giving an enlarged and more comprehensive vision about the type of studies that have addressed the relationship between a country's development and the dimension of sustainability more focused on.

There are many works in the literature pertaining to renewable energies and sustainability, but these topics are typically studied individually or use different research methods and target different goals. For instance, Picchi et al. [21] performed a literature review to explore the relationship between renewable energies and ecosystem services for landscape planning and design. Lammers and Hoppe [22], based on a literature review, investigated what local energy planning and implementation processes look like in the post-liberalization era. Jenniches [23] reviewed the literature for methods of assessing economic impacts of the transition to RE generation at the regional level. There are also some literature reviews on a specific source of RE, such as the work in [24], which explores the water pumping system, and the research in [25,26] looking at biomass for energy generation. Some research explores only one dimension of sustainability or specific sectors; for example, Sheikh et al. [27] investigated the social and political impacts of renewable energies and the authors of [28–30] focused on sustainability in a supply chain context, presenting it as a multi-disciplinary field with the contribution of many experts from a variety of areas.

It is also known that the use of RE has a positive impact on environmental sustainability. According to Franzitta et al. [31], the use of renewable energy sources, in particular wind, solar, biomass, and also sea waves, will reduce energy dependence on fossil fuels, reduce greenhouse gas (GHG) emissions, reduce environmental pollution, and improve the efficiency of the electrical grid. There is another source of RE which is the tidal power that besides contributes to reduce the dependence on fossil fuels it has a negative hydro-environmental impact since the tidal turbines alter ambient flow patterns because of the extraction of Kinetic energy [32].

Published research on RE(s) and sustainability has focused on different perspectives. For example, one topic receiving attention has been the influence on sustainability of a specific source of renewable energy: (i) Liu [33] focused on nuclear energy, proposing a system to assess the level of sustainability for a nuclear and renewable energy integration system employing a small modular reactor; (ii) Franzitta et al. [34] suggested generating electrical energy with a wave energy converter to reduce the production of electrical energy from traditional power plants and, as a result, their GHG; and (iii) Kyriakopoulos et al. [25] discussed biomass exploitation for electricity generation in a global-oriented and technological perspective reported in the literature. Some works also suggest different approaches for assessing the sustainable performance of renewable energy systems. For example, Wibowo and Grandhi [35] suggested constructing a performance index of renewable energy system choices using an algorithm assessing positive and negative ideal solutions. Considering the focus on RE(s), more emphasis on environmental and economic sustainability is expected, given the usually high costs associated with RE. In addition, Douziech et al. [36] argued that besides the tidal and waves energy plants have been considered as green technologies, since they do not alter the climate, conserve resources, have no harmful effect on human health or ecosystems, and are less harmful to the environment than conventional means of energy generations, an assessment of the amount of metal used by these technologies, however, shows an impact, respectively, 11 and 17 times higher than for coal- and gas-based power generators.

Investment in RE(s) is growing in almost all countries because of the important benefits that can result, mainly: lower greenhouse gas emissions and fossil fuel dependence, potential economic benefits, energy supply security [37], monetary benefits for neighboring communities and the entire region, and reliable energy supply [38]. However, some negative outcomes are also identified, such as environmental degradation [39], ecosystem disturbance, watershed damage, and noise and pollution during construction [37,40]. It is also known that energy supports not only the wealth growth but also the development of countries [41].

Works on renewable energies and sustainability cover a few topics, including systematic and quantitative assessment models of the sustainability of renewable energy [42], the study of the acceptance and impact of a specific renewable energy project (e.g., [43]), and a bibliometric analysis of a sustainability assessment of a specific RE. Unlike our study, however, these approaches present a narrow vision that does not contribute to an accurate perception of the state of the art of studies on the relationship between renewable energies and sustainability.

According to the current literature review, the following research questions are suggested:

RQ$_1$: Are there differences between the type of renewable energy focused on in the articles and the level of development of countries for which they are analyzed?

RQ$_2$: Are there differences between the dimension of sustainability focused on in the papers and the country's development level?

RQ$_3$: Are there differences between the type of renewable energies focused on and the dimension of sustainability analyzed in the articles?

As energy is a cornerstone of sustainable development and is the main challenge facing all countries, both developed and developing [44], it is valuable to determine the state of the art regarding renewable energies, sustainability, and the level of countries' development. The objectives of this paper are therefore the following: (i) to analyze the relationship between the type of renewable energy focused on in the papers and the level of development of countries analyzed therein; (ii) to identify the relationship between the dimension of sustainability focused on in the papers and the country's level of development; and (iii) to identify relationships between the type of renewable energies focused on in the papers and the dimension of sustainability analyzed. This review important for researchers who wish to identify hot topics in which research is lacking or that have already been investigated and for practitioners who wish to stay abreast of research on the topic.

The remainder of this article is structured as follows. Section 2 provides the background of the topic. Section 3 defines the methodology used, describes the data selection process, and reports an analysis of data from the sample of articles retrieved. Section 4 performs a more in-depth analysis to answer the research questions. Section 5 critically analyzes the results, highlighting the interplay between political discourses, policy, empirical evidence, and the findings of our review. Section 6 concludes and proposes questions for further research.

## 2. Background

Ness [45] introduced the model of economic development described as "take, make and dispose", whereby the exploitation of raw materials and non-renewable energy provided the basis of development of world economies, which in turn led to unprecedented growth. Unfortunately, this linear economic model highlights the economic goals at the expense of environmental and social dimensions, pushing the world to its physical limit. In fact, this linear model threatens the very stability of economies and the integrity of ecosystems that are vital for human survival. In this line, Yuan et al. [46] focused on the Chinese case and argued that the rapid economic growth of this country supported in the linear economic model has made the country a leading world economic power, increased the wealth of the population, and brought unprecedented business and employment opportunities. The downside is that all of this has provoked serious natural resource depletion and environmental pollution. In addition, recognizing the importance of China adopting a circular economy model, Feng and Yan [47] suggested implementing a framework to change the economic paradigm. Su et al. [48] pointed to environmental deterioration and scarcity of resources as two of the most urgent problems that must be tackled. They emphasized the importance of greater efficiency in the use of materials and energy to achieve a circular economy. Organizations find themselves compelled to implement strategies concerned simultaneously with the economic growth and sustainability as a way of addressing the challenges associated to the climate change, resource scarcity, dependence on fossil fuels, uncertainty in government regulations, high competitiveness, and globalization [49]. In this context, the pure economic business

perspective of companies is evolving to one that includes more regard for sustainability, adding social and environmental concerns to their operations as a result.

## 2.1. Level of Countries' Development and the Type of Renewable Energies

The publication of the 1972 Growth Limits [50] stimulated the international community to think about an alternative development model more concerned with sustainable global economic development, social progress, and environmental protection [51]. Sustainable development demands a collective effort to construct a future for society and the Earth that is both inclusive and sustainable. Together with businesses and social participants, governments are taking proactive measures to fulfill the UN's global sustainable development agenda by 2030 [52]. These ambitions are linked with many challenges, including the creation of new jobs, sustainable cities and industries, sustaining biodiversity, sustainable consumption and production, and addressing the challenge of climate change [53]. Energy is related and supports many sustainable development goals which are central to many of the challenges and opportunities facing the world and that are associated with income, pollution, and ecosystems.

Concerning energy demand and from the point of view in [54], developing nations differ from industrialized nations in both quantity and quality. As the standard of living increases, for example, there is greater demand for electricity in countries and in small, decentralized villages. Therefore, it would be greatly beneficial to electrify small communities of developing countries with alternative sources of energy from the outset in parallel with the rising standard of living in those communities. Gradually disseminating RE(s) to rural communities in a way that keeps pace with their development would bring considerable long-run benefits to their economies and environments.

As an illustrative tool, the authors of [12,55] described an energy ladder, which relates the energy mix with the level of a country's development. According to this energy ladder, richer countries tend to diversify the energy sources and abandon fossil fuel and hydroelectric sources to a greater extent, making them more dependent on sophisticated sources of RE (wind, solar, and manufactured biomass). This calls attention to the question of whether the relationship between RE and sustainability mirrors the energy ladder or some other relationship between the energy mix and the level of countries' development.

## 2.2. The Level of Countries' Development and the Sustainability Dimensions

As mentioned above, sustainability has three dimensions: economic, social, and environmental. For example, Yamaguchi [16] highlighted the Japanese Official Development Assistance program, which joins a central strategy addressing environmental sustainability and economic growth. This policy gained visibility at the 1989 Arche Summit, when Japan committed to expanding its contributions to the environmental field. According to Yamaguchi [16], energy problems represent a worldwide issue being closely related to the response to global environmental problems and the achievement of sustainable development.

Another example is the case of Brazil, which, according to the World Bank Classification, is considered a country with an upper-middle level of development and is the largest soybean exporter in the world [56]. The growing trade in soybeans has contributed to the deforestation of the Amazon region [57]. This has spurred migration into the area, causing land grabbing and land speculation, which exacerbate social conflicts [58]. Deforestation, especially of the Amazon rain forest, has increased GHG emissions and accelerated biodiversity loss [59]. Because of these threats, the effectiveness of environmental policies that protect the Amazon region is crucial [60]. The relationships between sustainability and countries' development is thus an important topic contributing to a better understanding of whether countries with different levels of development also have different sustainability behaviors.

*2.3. Relationship between Renewable Energies and Sustainability*

Much of society acknowledges the key role of energy in supporting sustainability goals. This is especially recognized in the case of RE, along with growing attention to the benefits that it can offer to achieve the "Sustainable Energy for All" goals, to reduce poverty, boost economic growth, and in general promote sustainable development [61,62]. This justifies the importance of examining published papers to see the extent to which the relationship between renewable energies and sustainability has been addressed.

The influence of renewable energies on sustainability has been the focus of some works, but mostly separately. Some literature reviews on RE(s) can be found with a special focus on methods of assessing regional economic impacts of a transition to RE generation [23], with the main objective of characterizing local energy planning and implementation processes in the post-liberalization era [22], or to determine the social and political impact of RE(s) [27]. In addition, the research developments with renewable energy source water pumping systems are reviewed in [24]. Renewable energy in the service sector in general and in the tourism industry specifically is explored in [28]. A literature review on the factors that affect the performance and growth of clean technology start-up firms was performed by Bjornaly and Ellingsen [63].

## 3. Method and Data

*3.1. Method*

A bibliometric analysis was performed, seeking a better understanding of the directions of scientific trends concerning the relationship between renewable energies and sustainability. Bibliometric analysis looks at publications and their properties [64] and adds knowledge domain visualization to gain a sense of the development and evolution of a knowledge field.

The quality of a bibliometric analysis depends on the quality of the input data, and it is essential to approach the literature in an unbiased way. The Scopus database for the period 1997–2019 was used to collect data. This time range was chosen because 1997 was the period when the literature on renewable energies and sustainability began to grow (see Figure 1). Our period of analysis finishes in August 2019, the month in which the data were collected.

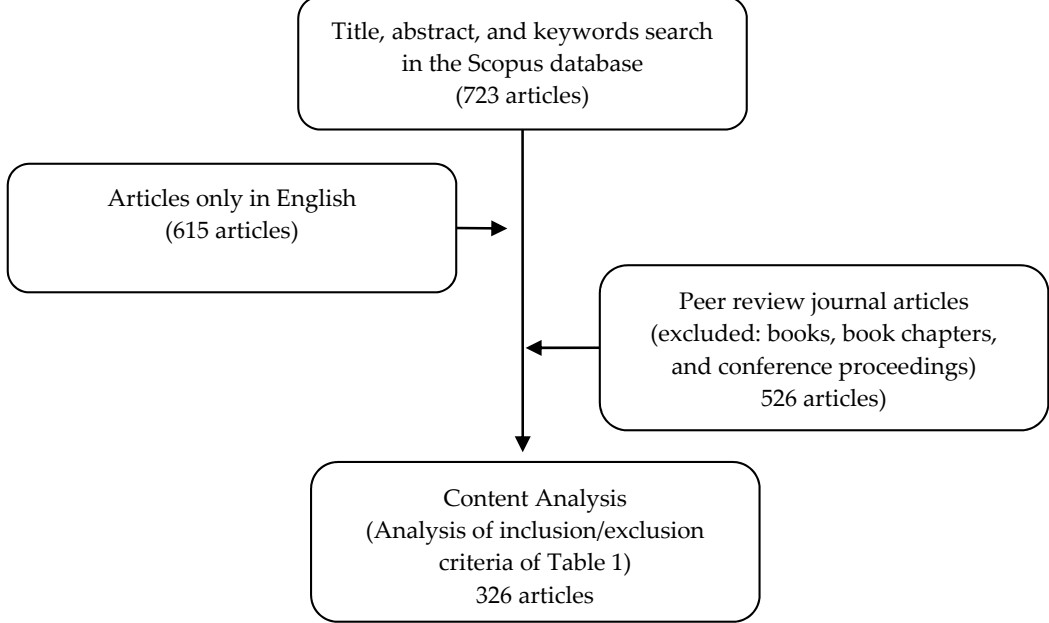

**Figure 1.** Literature search.

Bibliometric analysis relies on quantitative methods to investigate a body of publications. The bibliometric indicators used herein include the total of publications on the research topic during the period, the main journals in which the research is published, the scientific fields in which the topic has been covered, the number of citations by journals, the network analysis, and keywords occurrence. The visualization software VOSviewer (v1.6.10, Centre for Science and Technology Studies CWTS, Leiden, The Netherlands) was used, which is a science mapping method of bibliometric analysis [65].

The four-step methodology suggested by Zhao and Strotmann [66] was followed in this study: (i) define the search keywords; (ii) clean and format the data; (iii) make an initial analysis; and (iv) perform the final data analysis.

Regarding Step (i), using the approach suggested by Fahimnia et al. [67], the title, abstract, and keywords were searched in the Scopus database (Elsevier, Amsterdam, Netherland). The Scopus database (Elsevier) was selected because it is the largest database of abstracts and citations for scientific peer-review literature and includes more than 22,000 titles from international publishers.

Keywords searched in title and abstract were renewable energy, its synonyms (renewable resources, bioenergy, and bio-energy) and the following renewable energy types: biomass, biofuel, biodiesel, ethanol, bioethanol, wind energy, wind power, solar energy, solar power, photovoltaic cells, geothermal energy, heat energy, ocean-power, hydropower, water-energy, water power, hydroelectric power, and landfill gas. All these keywords were included in a non-exclusive way (with the "OR" particle) to identify all the articles covering all types of renewable energies. Additionally, the keyword sustainability (social, environmental, economic sustainability) was also included to those keywords be presented in the search fields (title, abstract, keywords) in each publication. This procedure made possible to identify the publications that cover not only the topic of renewable energies but also some dimension of sustainability.

In Step (ii), the search was limited to articles not only written in English, but also published in indexed journals that are subject to the peer review process. Books, book chapters, and conference proceedings were not considered since not all are subject to peer review practices. This stage identified 526 articles. To use VOSviewer, the data were standardized and formatted to a plain text making it possible to analyze their relevance (see Table 1). A total of 316 records adhered to these filtering criteria.

**Table 1.** Appraisal Step (ii): inclusion and exclusion criteria.

| Inclusion/Exclusion Criteria | Rationale |
|---|---|
| 1. The article must demonstrate that the adoption of renewable energies must be made by the supply side of the economy (namely, but not exclusively, firms) | As the research is not restricted to specific journals, research on units of analysis other than the ones from the supply side may occur, namely adoption by the demand side (e.g., families). |
| 2. The article must focus on at least one dimension of sustainability. | Considering the wide search parameters, some articles may approach sustainability as a secondary issue. |
| 3. The article must focus on both the use of renewable energies and sustainability. | Considering the wide search parameters, some articles may approach potential relationships in hypothetical terms. |
| 4. The article must be based on quantitative or qualitative analysis, or a mixture of both methods | Conceptual articles that do not study a relationship between renewable energies and sustainability are not analyzed |

A summary of the bibliographic search is presented in Figure 1.

In the next steps, an analytical framework was developed to identify some patterns in the research field. The articles extracted in the second phase were used as the input data and processed using the *Excel* (Version 2019 (16.0), Windows, Microsoft, Redmond, Washington, EUA) and *VosViewer* softwares (v1.6.10, Centre for Science and Technology Studies CWTS, Leiden, The Netherlands).

In Step (iii) (developed in Section 4), the time trend of publications and their citations were analyzed, classifying them into different subfields, identifying the major outlets in terms of quality impact factors, establishing a keyword network, and characterizing the academic community contributing to this literature.

Finally, Step (iv) sought to attach meaning to the patterns and data revealed during the synthesis step. Sense making and interpretation help to understand how arguments, interests, and research questions have evolved over time and what the focus in the future will tend to be. By doing this, an indication of the hot topics covered by these researchers is reached and the additional emerging research fields are identified [67]. It is also in this stage that our research questions were answered.

### 3.2. An Overview

The literature on renewable energies and sustainability has been growing since 1997, showing an explosive path from 2004 to August 2019, reaching a peak in 2012 with the publication of 41 documents (see Figure 2).

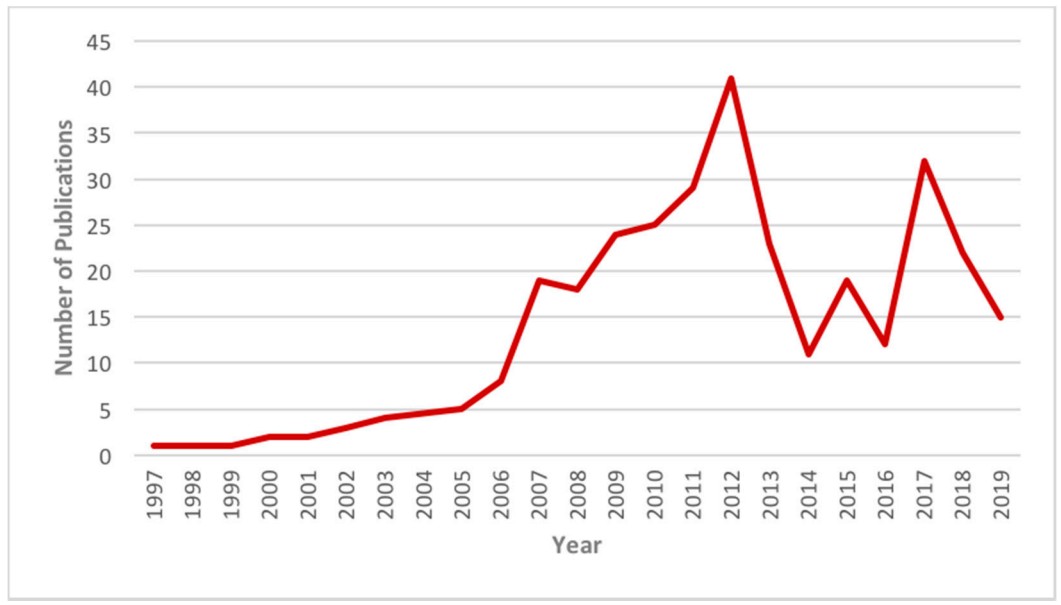

**Figure 2.** Number of articles published on renewable energies and sustainability.

In Figure 2, it is important to note that the number of articles published on the research topic during 2019 includes only the first eight months (until August) of that year.

### 3.3. Subject Classification of Publications

The articles selected were published in a great range of journals assigned to one or more subject categories. Figure 3 shows the top active subject categories of renewable energies and sustainability publications based on the classification of subject categories in the SCOPUS database. The publication output for renewable energies and sustainability research is distributed into 10 subject categories. The three most active categories are Environmental Science (23% of scientific papers), Energy (18% of scientific papers), and Engineering (13% of scientific papers). Other active subjects include Business, Management, Accounting, Social Sciences, Agricultural and Biological Sciences, Material Science, Chemical Engineering, Economics, Econometrics, and Finance.

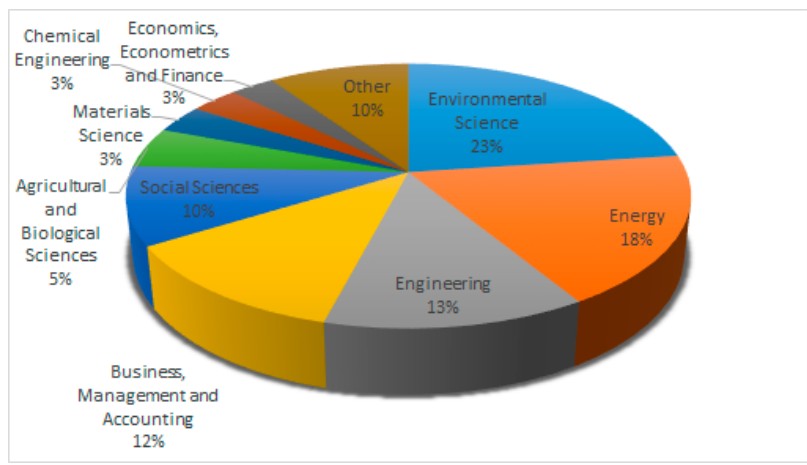

**Figure 3.** Top active subjects in renewable energies and sustainability publication output.

*3.4. Network Analysis—Keywords Co-Occurrence*

The "hot topics" in a specific field can be defined by high-frequency keywords in an appropriate database. The keywords summarize the content of a research article, and serve to focus and refine the main ideas of the research [68].

For bibliometric application, Figure 4 shows the top keywords in renewable energies and sustainability papers.

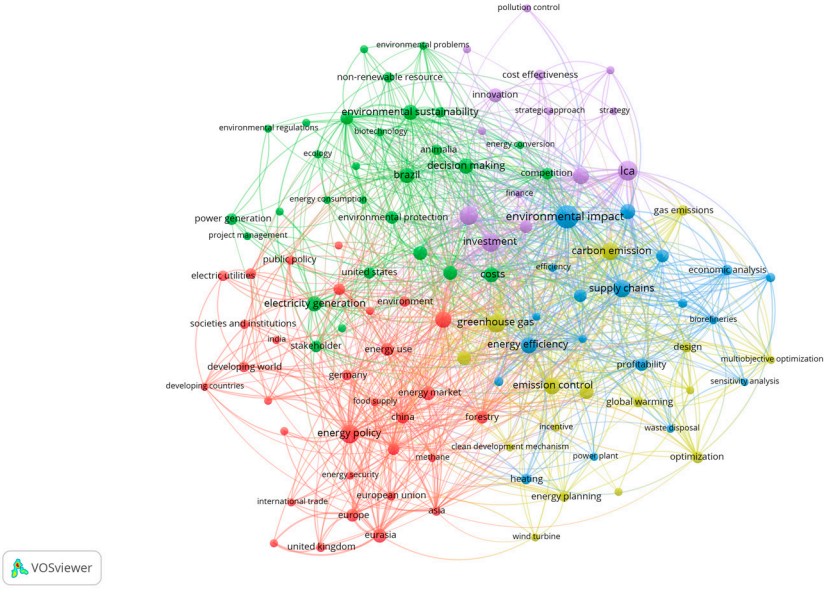

**Figure 4.** Network of co-occurrence keywords.

In this figure, each term is represented by a circle and its size indicates the number of publications that include that term in the title or abstract. Terms that often co-occur tend to be located close together in the visualization. The terms are grouped into five clusters by using VOSviewer, of which four are of significant size. To obtain an unbiased result, keywords that somehow were identical or already used in the search in the Scopus database were excluded (the keywords excluded were: "sustainability", "sustainable development", "renewable resource", "biofuel", "biomass", "biofuels", "renewable energy resources", "bioenergy", "alternative energy", "environmental sustainability", "renewable energies", "renewable energy", "fossil fuels", "biodiesel", "bio-energy", "biomass power", "fossil fuel", "industry", "solar energy", "wind power", "ethanol", "renewable energy source", "sustainable energy", "solar power", "biogas", "sustainability assessment", "photovoltaic system", "hydroelectric

power", "fuels", "environmental performance", "biofuel production", "algae", "photovoltaic cells", "economic sustainability", "renewable resources", "energy", and "geothermal energy"). After all the requirements were satisfied, VOSviewer generated the network with the next five clusters, which were carefully analyzed and defined with different classifications: "Pollution effects and control" (yellow), "Policies and development" (red), "Innovation and production" (purple), "Environment" (green), and "Economic efficiency" (blue). The main keywords considered as making up part of the cluster "Pollution effects and control" are: emission control, carbon emission, greenhouse gas, global warming, gas emissions; energy conservation, economic effects, and social effects. Grouped into the cluster "Policies and development" are the main keywords: energy policy, energy market, Eurasia, Europe, energy use, and developing world. Looking at the cluster named "Innovation and Production", some of the keywords that can be found are: investment, LCA, competition, innovation, and cost effectiveness. In the group named "Environment", the following main keywords can be found: environmental sustainability, decision making, Brazil, electricity generation, biotechnology. Finally, in the cluster "Economic efficiency", the main keywords are: environmental impact, energy efficiency, bio-refineries, economic analysis, supply chains, profitability, and efficiency.

The network analysis using the keywords occurrence, visualized in Figure 4, presents the focus and trends of the selected studies on renewable energies and sustainability. There are 2864 keywords in total. Of these keywords, 632 repeated twice, 333 repeated at least three times, 213 keywords at least four times, and 147 keywords appeared five times.

The network suggests that the biggest node is "environmental impact", which occurs 38 times and is connected to every single cluster, with 79 different links. It is followed by "LCA" (life cycle assessment) with 28 occurrences; "climate change", "energy policy", "greenhouse gas", and "investment" with 25 occurrences; and "supply chains" and "carbon emission" with 24 occurrences. According to the network analysis, it can be concluded that most works on renewable energies and sustainability focus mainly on environmental problems and attempt to find solutions regarding policies in the supply chain and investments of the economy.

The clusters and keywords identified highlight a major concern with the environmental and economic dimensions of sustainability, just after the social dimension of sustainability. It is important to note the role of innovation in energy production (or production in general) in the literature studied in this article. It is also worth noting that most of the studies deal with industrial sectors and only a few with agriculture, services, and tourism. Moreover, the importance of studies concerning the biggest developing countries in the emergence of this literature is highlighted by bullets related to China, United Kingdom, India, Germany, and Brazil with a special focus on the policies and development processes. Notwithstanding, some developed countries or groups of countries (USA and Brazil) appear within the "Environment" cluster.

*3.5. Most Productive Countries on Renewable Energies and Sustainability*

In this section, the most productive countries in terms of renewable energies and sustainability concerns are identified and the temporal distribution of the sample articles, enhancing the main research topics focused on during the focused period are analyzed.

3.5.1. Spatial Distribution of Most Productive Countries Producing Renewable Energy and Having Sustainability Concerns Identified in the Articles

The contributions made by different countries to the research topic are estimated by using the location of the renewable energies investigated by the articles. This means that the most productive countries are those in which more articles about renewable energies (implanted in those countries) and sustainability concerns are identified. The renewable energies and sustainability topic cover 55 countries. The most-active countries are identified in Figure 5, in which the most representative are seen to be the following: USA (13.04%), Brazil (12.08%), EU (European Union) (5.28%), China (5.80%), and UK (5.80%). The analysis of the country contribution reveals that the top 5 most active countries,

not considering the "others", accounting for 43% of all publication outputs, are USA, Brazil, EU, China, and UK. The "others" category has all the countries that have only one or two articles on the research topic.

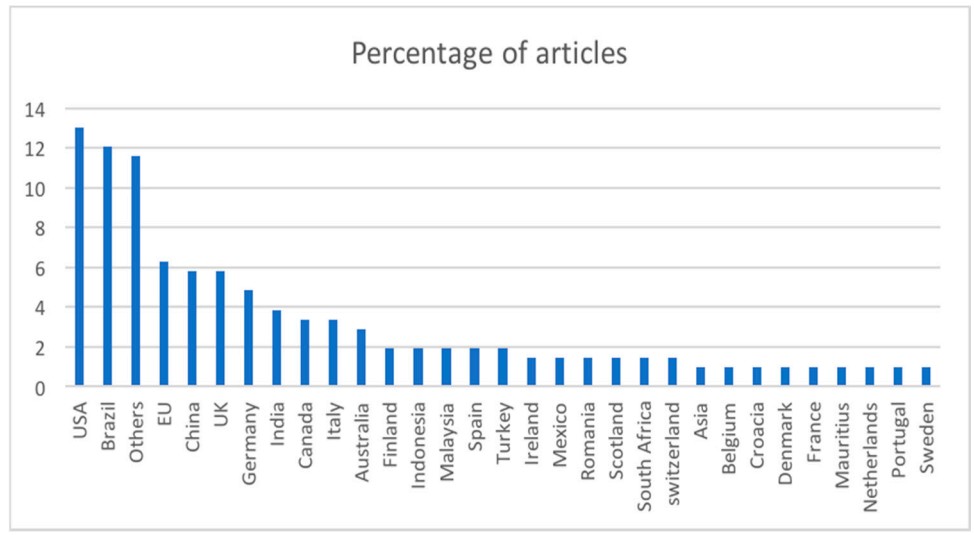

**Figure 5.** Most productive countries producing renewable energy and sustainability articles.

### 3.5.2. Temporal Distribution of Most Productive Countries of Renewable Energies and Sustainability Articles

Figure 6 is a visual presentation of the top 10 countries' affiliated institutions having at least one author of each paper published from 1997 to August 2019. It is noticeable that the publication outputs in this topic have increased dramatically since 2005. This growth in renewable energies and sustainability publications since this date could reflect the World Summit on Sustainable Development that took place in Johannesburg in 2002, calling for commitment to "encourage and promote the development of renewable energy sources to accelerate the shift towards sustainable consumption and production" [69], which prompted researchers around the world to publish papers on the issue of renewable energies and sustainability. For example, Omer [70], from the UK, published a work calling for innovative renewable applications and the need for reinforcing the renewable energy market in order to better preserve the ecosystem by reducing emissions and to improve environmental conditions by replacing conventional fuels with renewable energies that produce no air pollution or greenhouse gases. In addition, Mores et al. [71], whose main author affiliation is with Brazil, analyzed the level of innovation in the production of green plastic by using ethanol made from sugarcane, which is a renewable resource, instead of naphtha, which is considered a non-renewable resource in the context of sustainable supply chain.

Over time, the topic of renewable energies and sustainability has taken different directions and concerns. In 2009, concerns on performance assessment of renewable energies options and sustainability (environmental concerns) was the main scope of much of the research in this field developed by researchers from UK, USA, and Germany. For example, Pearce et al. [72] provided a graphic tool to determine the return on investment of any energy conservation to encourage the increased deployment of energy efficiency and renewable energy technologies. In addition, Pigaht et al. [73] addressed innovative private micro-hydro power projects in Rwanda, exploring the real impact on performance and sustainability and enhancing the importance of the existence of a true collaboration of local private and financial sector firms. Halog [74] developed integrative operations-based metrics considering the multi-expectations of various stakeholders to allow for a sustainable development of biofuels production supply chains.

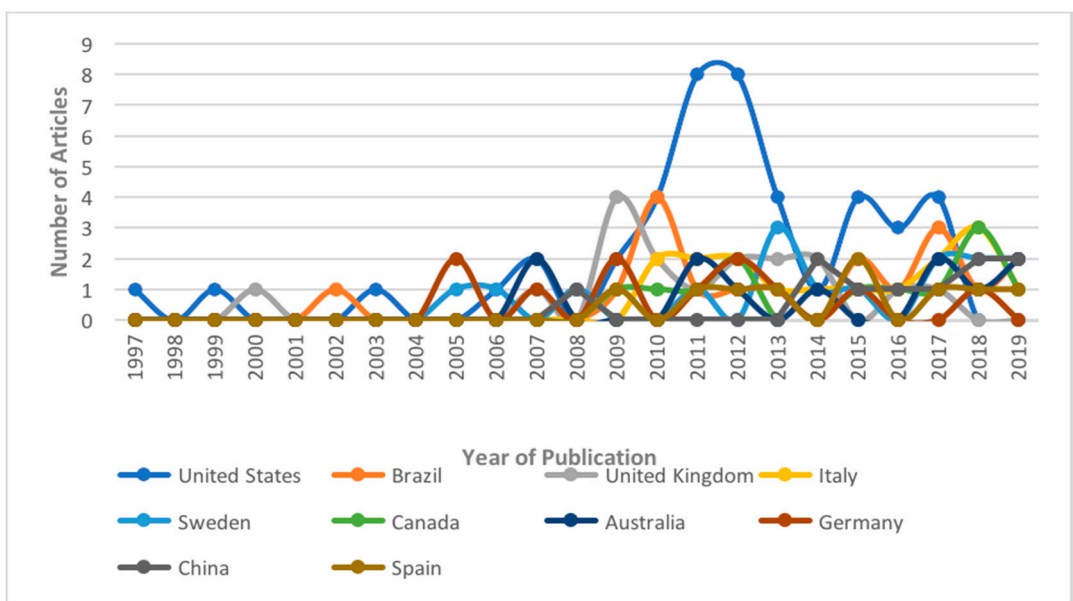

**Figure 6.** Temporal distribution of top 10 most productive countries of renewable energies and sustainability articles.

In 2010, researchers affiliated with institutions from the USA, Brazil, Italy, and Canada developed articles focusing on specific types of renewable energies and their sources making different contributions to this research topic. For example, Danon et al. [75] focused on wood biomass as a source of energy production, giving insights on the main factors affecting the sustainability of future commercialization of wood residue such as the availability of the wood raw material, the development of wood-based fuel markets, and expectations related to the profit. In addition, Neves [76] listed a set of actions for companies and governments to be more sustainable promoting a discussion of ethanol, which is considered one of the most viable clean and renewable fuels used by society until the present. Espinoza and Vredenburg [77] focused on wind, arguing that environmental, institutional, and cultural factors play important roles in the emergence of wind energy industries in both industrialized and emerging economies. Biopolymers were evaluated by Chadha [78] in terms of their chances and risks, highlighting the importance that biopolymers have gained in the industry and describing the different strategies that firms apply to employ biopolymer technology successfully.

Many of the publications on renewable energies and sustainability during 2011 have a special focus on sustainability models [79–81] and sustainability assessment methodologies associated with renewable energies [82–84]. The affiliation of their authors are mainly institutions in USA (Figure 6).

The year 2012 was very productive in terms of publications, with a specific set of topics being explored by a considerable number of authors affiliated with USA, Italy, Germany, and Spain, such as biofuel/bioenergy [85–87], benchmarking analysis and case studies on renewable energies [88–90], energy policies and governance [91,92], and energetic analysis of renewable energies [93,94].

Many articles on renewable energies and sustainability are observed also during 2017 and with different research topics. The state of the art of specific renewable energies in some countries was explored in some articles published in this year, such as biomass energy in Turkey [95] and Malaysia [96]; biofuel in Malaysia [97], Australia [98], and Sardinia [99]; hydropower in India [100]; and the energy sector in general in Romania [101]. The policies associated with renewable energies have also been explored in some articles: (i) Leoneti et al. [102] focused on the policies related to renewable energy resources regarding their industrial processes, the role of government incentives or subsidies, and investments of companies in technology development; (ii) Tagotra [103] highlighted policies associated with the renewable energy sector in India post-Paris negotiations; and (iii) the authors of [104] underlined the misguided goals and inefficient mechanisms of biofuel policies. The economic

and financial performance of the renewable energy sector has also been the focus of some of the sample articles (e.g., [101,105,106]). From these works, it is interesting to highlight the main conclusion drawn by Paun [101], who considered that the investments in renewable energies have been considerably opportunistic, based on the wish of the government to maintain the subsidies it introduced, instead of being based on the realistic long-term financial performance of the companies in this area. It is also interesting to note that the research unit focused on in the sample articles has changed somewhat over the years. In fact, the number of articles on supply chain in the renewable energy sector has showed substantial growth. A focus on the supply chain can be found in [107] about waste paint in auto industries, in [108] focusing on the solar cell industry, and in biodiesel production using waste cooking oil [109].

Pertaining to 2019, and considering only the first eight months, the countries where more research has been performed on the topic are China and Spain. For example, Yu et al. [110] assessed the sustainability of renewable energy development and use in China's 30 provinces from 2011 to 2015.

## 4. The Most Important Topics and Analysis

*4.1. Level of Countries' Development Focused on in the Papers and the Type of Renewable Energies Analyzed*

This section investigates whether there is a relationship between the level of countries' development and the type of renewable energies that are addressed in the research articles (RQ$_1$). First, however, the most explored sources of renewable energies focused on in the sample articles are analyzed.

Several types of renewable energies, with many of the energy sources being approached either simultaneously or alone are investigated in the research articles, such as biomass, biofuel, multiple (several renewable energies are focused on simultaneously in the articles), solar, biodiesel, ethanol, and hydroelectric. Biopolymer and hydrogen are also identified in the sample, but only barely. Table 2 shows the subsample of articles focusing on only one source of renewable energy.

As shown in Table 2, biofuel is the type of renewable energy most focused on in the research articles, followed by biomass. After biomass, solar, hydroelectric, ethanol, and biodiesel are the sources of energy most focused on in the sample articles.

The level of a country's development focused on in the papers and the type of renewable energies are also reported in Figure 7.

As shown in Figure 7, the main sources of RE focused on in articles developed in high-income countries (Austria, Canada, Finland, Italy, USA, etc.) are biomass [111–115], wind–solar [116–118], and biodiesel [104,110,119]. Although the same renewable energies are focused on in countries with the same level of development, the research objectives are quite different. For example, in [116], wind–solar is focused on with the objective of suggesting an energy model with the main concern of improving economic and environmental performance; in [117], an approach is suggested to evaluate the profitability of the solar energy system in a context of self-consumption; and, in [118], the competitive market of North Carolina between large solar power producers and utility companies to finance, install, and operate solar generating systems is analyzed. Considering biomass, Menrad et al. [113] focused on policy and legislation conditions as mandatory to establish stable conditions and provided planned security for investment decisions on biomass exploitation, and Plieninger et al. [114] identified pitfalls impeding a broad implementation of wood energy supply in forestry.

Considering the works about biodiesel, while Oliveira et al. [104] focused on policy mechanisms to facilitate more socially and environmentally sustainable energy production, Joensuu and Sinkko [119] explored sustainability and improvement options for biodiesel supply chains.

**Table 2.** Main renewable energies focused on in the sample articles.

| | Biomass | Biofuel | Solar | Biodiesel | Ethanol | Hydroelectric |
|---|---|---|---|---|---|---|
| 1997 | | | | | | |
| 1998 | | | | | | |
| 1999 | | | | | | |
| 2000 | | | | | | |
| 2001 | | | | | | 1 |
| 2002 | | | 1 | | | |
| 2003 | | | | | | |
| 2004 | | | | | | |
| 2005 | 1 | | 1 | | | |
| 2006 | | 1 | 1 | | | |
| 2007 | 1 | | 1 | | | 1 |
| 2008 | 1 | 1 | | | | |
| 2009 | | 3 | | | 1 | 1 |
| 2010 | 1 | 3 | | | 2 | |
| 2011 | 2 | 2 | 2 | 2 | 1 | |
| 2012 | 1 | 6 | 3 | 2 | 1 | |
| 2013 | 1 | | 1 | | 3 | 3 |
| 2014 | | 1 | 1 | | | 1 |
| 2015 | 3 | 1 | 1 | | | 1 |
| 2016 | | 1 | | 2 | | |
| 2017 | 3 | 4 | 2 | 3 | 2 | 1 |
| 2018 | 4 | 2 | 1 | | | 2 |
| 2019 | 7 | 4 | | | | |
| Total | 25 | 29 | 15 | 9 | 10 | 11 |

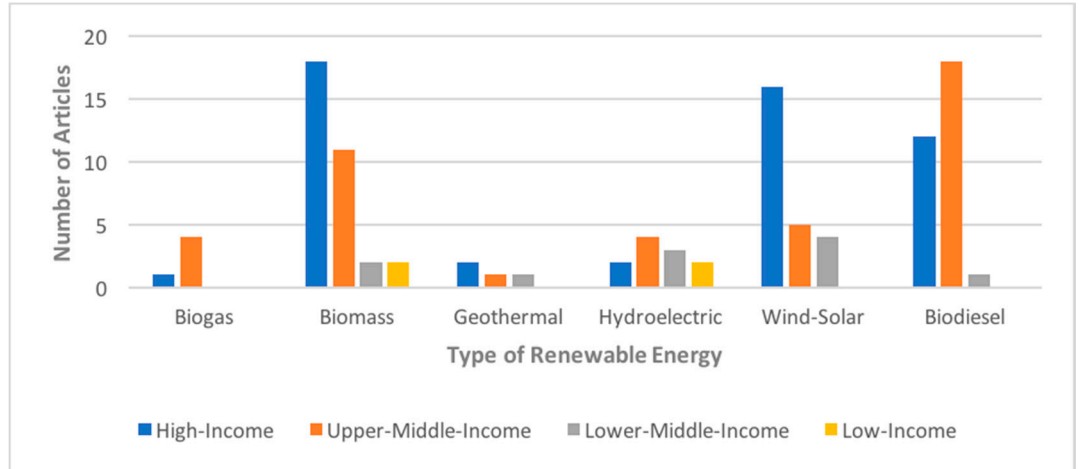

**Figure 7.** Type of renewable energies by level of countries' development analyzed in the research sample.

In papers focusing on upper-middle-income countries (China, Brazil, Turkey, Malaysia, Colombia, etc.), the type of renewable energy most studied is biodiesel [109,120,121], and the least studied is biogas [122,123], but with different aims. In studies about biodiesel, supply chain analysis seems to

be a critical issue and the idea that firms focusing on individual sustainable development elements independently are unlikely to find satisfactory solutions to their sustainability problems associated to biodiesel's SC seems to be common [109,120]. In addition, Bautista et al. [121] proposed a model to assess the sustainability of the conditions in the baseline scenario of biodiesel production.

Biomass and hydroelectric are the only two types of RE that have been focused on in the sample articles independently of the development level of the focused countries. Moreover, the upper-middle-income and high-income countries are also those in which interest is most focused on renewable energies. According to the analysis of the sample articles focusing on high-income countries, several common topics can be identified: (i) energy policies [85,124–130]; (ii) energy efficiency [99,131–139]; and (iii) technological innovations [93,112,140–143].

In articles focusing on lower-middle-income countries, besides energy efficiency [144,145] and energy policy [100,103], corporate strategy [73] seems also to be a hot topic.

The results derived from this analysis are in line with those reported in the literature. According to the authors of [146] and [115], investment in renewable energies depends on the type of economy in which countries are operating. There are countries that are still operating in the "brown economy", which excludes sustainable development, and others that have made the transition to the "green economy", which "results in improved human well-being and social equity, while significantly reducing environmental risks and ecological scarcities" [147]. From the perspective of the authors of [148] and [149], the concept of green economy has gained great attention because of several important crises confronting the world in the beginning of the 21st century, such as the financial and economic crisis of 2008, crises in climate, biodiversity, food, fuel, and water. Although the causes of these crises may differ, UNEP [148] highlighted the misallocation of capital, as the most important being necessary to change the focus on a great investment in renewable sources of energy.

This resembles the "energy ladder", i.e., the relationship between development and the energy mix, which shows evidence that the process of development leads to a progressive replacement of fossil fuel and hydroelectric sources (and basic biomass) by a more diversified mix that includes solar, wind, waves, and transformed biomass [12,55].

In fact, in the articles developed in high and upper-middle income countries, the variety of renewable energies focused on is greater, since those countries invest more in key sectors of the green economy. This helps to decouple economic growth from its environmental impacts and the use of resources by shifting from non-renewable to renewable sources of energy.

Based on the results reported in articles focusing on high-income countries, the renewable sources of energies most studied are biomass, wind–solar, and biodiesel. In upper-middle-income countries, the type of renewable energy most studied is biodiesel, and the least studied is biogas. Biomass and hydroelectric sources of energy reveal no relationship with the development level of the countries.

*4.2. Relationship between the Sustainability Dimensions and the Level of Countries' Development*

This analysis seeks to know if there is a relationship between the dimension of sustainability focused on in the sample articles and the development level of countries (RQ$_2$) (Figure 8). Two criteria were used to identify the dimensions of sustainability focused on in the sample:

(i)   presence of the following phrases in the title, abstract, or keywords: "economic sustainability", "social sustainability", "environmental sustainability"; and

(ii)  mention in the article's title of the word "sustainability" and in the abstract or keywords some practices within each dimension of sustainability.

As shown in Figure 8, in the articles focusing on high-income countries (i.e., Australia, Canada, Finland, Italy, USA, Denmark, Netherlands, Sweden, Germany, etc.), the three dimensions of sustainability are explored, mostly on the environmental dimension of sustainability [40,126,136,150–152]. Additionally, in the articles developed about low-income countries (Rwanda, Senegal, Ethiopia, Indonesia, etc.), the topic of renewable energies and sustainability has

been less explored. Nevertheless, in the few articles developed in these countries, environmental sustainability is also the most explored [153–155]. Moreover, considering all the articles that comprise our sample, the environmental dimension of sustainability has been the most studied, followed by the economic one.

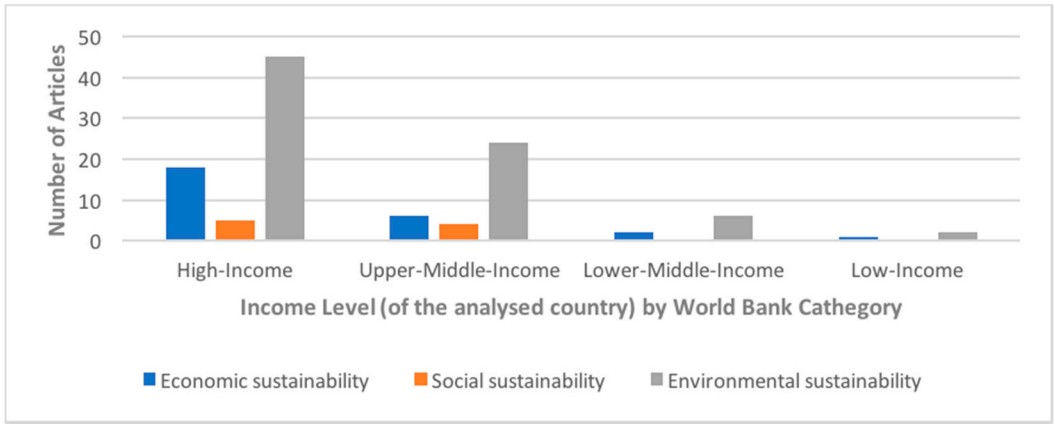

**Figure 8.** Dimensions of sustainability focused on in articles by development level of countries.

It is also interesting to note that the social dimension of sustainability is explored mainly in articles focusing on high and upper-middle income countries [124,156]. Summing up, the higher is the developmental level of the country, the more concern there is with the environmental dimension of sustainability. The same is observed in our sample regarding the economic dimension of sustainability, but with much less frequency (see [40,111,117,157–161]). This is consistent with the keyword co-occurrence network shown in Figure 4, where social concerns did not explicitly emerge.

According to the analysis, there is no clear relationship between the dimension of sustainability focused on and the development level of countries. It is observed that the environmental dimension is always the most focused on (followed by the economic one), regardless of the development level of the country.

Summing up, in the articles focusing on high-income countries, the three dimensions of sustainability are explored, but with a higher predominance of the environmental dimension of sustainability. Contrarily, in the low-income countries, the topic of renewable energies and sustainability has been less explored, but the environmental sustainability is still the most explored. Moreover, in our sample, the environmental dimension of sustainability has been the most studied, followed by the economic dimension, but with less frequency. The social dimension of sustainability is studied only in articles focusing on high and upper-middle income countries. In other words, the higher is the developmental level of the country, the more concern there is with the environmental dimension of sustainability.

### 4.3. Relationship between Renewable Energies and Sustainability

In this section, the relationship between the type of renewable energies focused on in the articles and the dimension(s) of sustainability (RQ$_3$) is explored (Figure 9).

As shown in Figure 9, wind–solar is the type of renewable energy most focused on in the articles, and the dimension of sustainability most explored in these works is the environmental one, either alone as in [108,162,163] or together with the economic dimension of sustainability as in [103,116,129]. This same result is observed with biomass, but to a lesser extent since the number of articles focusing on only the environmental sustainability is low [26,115,164].

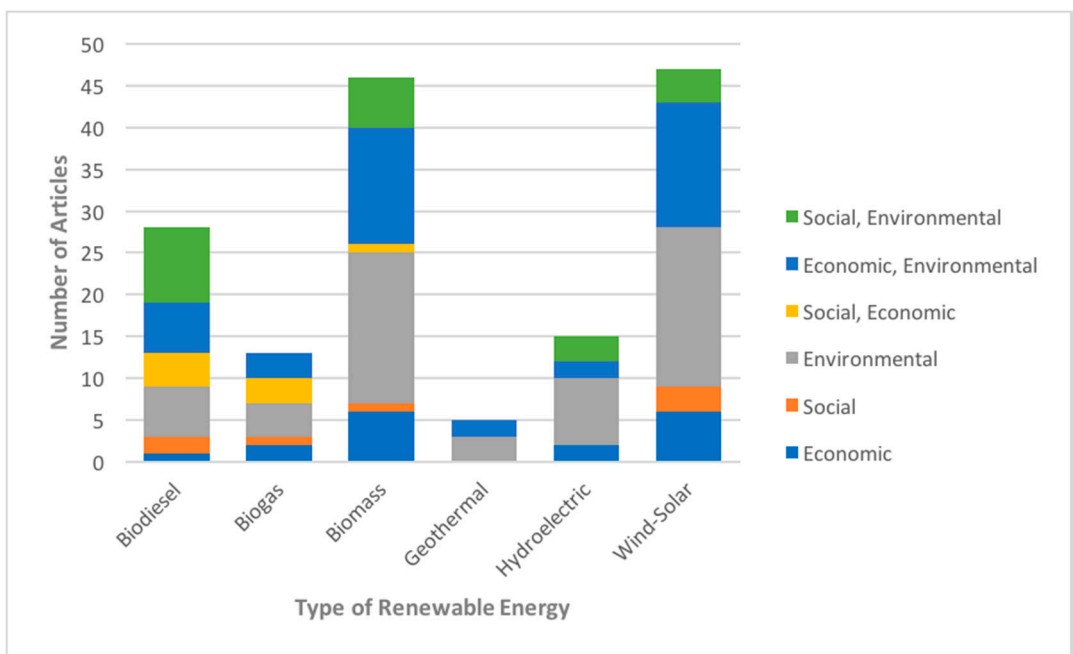

**Figure 9.** Renewable energies and the dimension of sustainability focused on.

As noted above, Figure 9 underscores that the environmental sustainability alone or with other sustainability dimensions is the most focused on. The social dimension of sustainability is the least focused on, as it has become a topic of investigation only more recently (see [71,125,165–167]). In the sample articles, the social dimension of sustainability is not mentioned in either geothermal or hydroelectric energy articles.

Biodiesel is clearly the exception to the pattern. In the articles focusing on this source of energy [120,168,169], the social and economic dimension of sustainability prevails over the environmental dimension.

The results of the study make it possible to state that there are differences between the type of renewable energies focused on and the dimension of sustainability analyzed in the articles examined. The wind–solar source of renewable energy is the most focused on, followed closely by biomass with the same behavior, with the environmental dimension of sustainability being the most explored.

## 5. Further Thoughts and Critical Analysis

The results reached with this analysis reveal a relationship between sustainability concerns and the countries' development level resembling the energy ladder. Despite an increase in the number of scientific publications on possibilities of prosperity without growth [170], some international strategic documents, such as the Green Growth Strategy [171], European Union Strategy 2020 [172], and "The Future We Want" [173], as well as the mainstream economics, still see economic growth (countries' development level) as a required component of sustainable development. Currently, some world leaders (e.g., the President of the USA, Donald Trump) dismiss the problem of global warming in the public discourse, which seems to discourage public and political emphasis on sustainability.

Some researchers state that a possible slowdown in a country's growth (see, e.g., [174,175]) may cause a reduction in investments in expensive cleaner technologies and consequently an increase of environmental and social pressure [176,177].

In this article, a review of literature relating renewable energies (adoption) from the supply side of the economy and the concept of sustainability is performed, including its three different dimensions: economic, environmental, and social. To that end, a set of tools from bibliometric analysis was used and a thorough analysis of the trends in the literature performed, finding that this literature is more concerned with environmental aspects of sustainability and mimics the empirical fact related

to the energy ladder with more diversified modern renewable energies studies undertaken in more developed countries. It is important to discuss the results of this research in the light of the political and empirical evidence.

Figure 10 summarizes the highlights of this critical discussion.

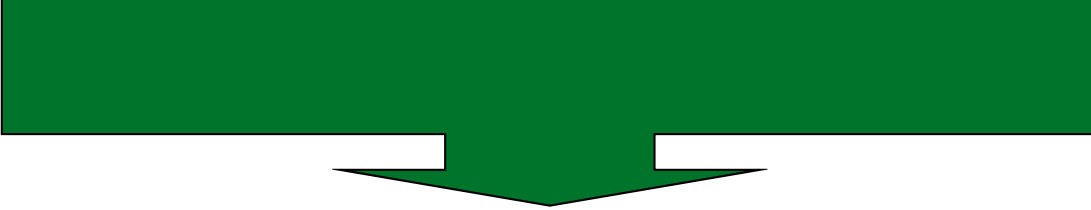

## Sustainability and Renewable Energies

| **Political and Institutional Discourse** | **Empirical Evidence** | **Literature Review** |
| --- | --- | --- |
| Mixed, with recent increase in powerful political discourse against the environmental dimension of sustainability | Energy Ladder | Mixed Sources of Energy analyzed in richer countries, with solar-wind energies being the major contributors for the analysis |
| UN Sustainable Development Goals (SDG) focusing on the three dimensions of sustainability: <u>Social</u> - (1) decreasing hunger, (2) poverty and (10) inequality); <u>Economic</u> - (8) economic growth; <u>Environmental</u> - (13) climate action and (7) clean energy. | Developed Countries based on diversified renewable energy resources<br><br>Developing countries still based on fossil fuels (and some on raw biomass) | Environmental concerns are overwhelmingly represented<br><br>Social concerns are almost neglected<br><br>On the contrary to what happens with other sources, articles focusing on Biodiesel are also more concerned with the econ |

- Dissociation between public discourse and the UN SDG;
- Dissociation between the UN SDG and literature that almost neglects the social dimension;
- Empirical evidence reveals a hopeful trend mostly due to development;
- Important focus on innovation processes in the literature;
- Need to accelerate research on modern renewable energies in developing (and poor) countries and on the social dimension in order to accelerate the transition through the energy ladder and help to fulfil the SDG.

**Figure 10.** Summary of the critical analysis relating the findings of the article with political discourse and empirical evidence.

On the side of political agenda, there is a dichotomy between political discourses of some world leaders against the environmental concerns (namely, related to global warming) (some newspaper references on the issue are on Vox (https://www.vox.com/2019/2/5/18207337/state-of-the-union-2019-climate-change) and BBC (https://www.bbc.com/news/world-us-canada-45859325)) and the United Nations Sustainable Development Goals (https://sustainabledevelopment.un.org/?menu=1300), which focus on climate change mitigation, green economy, and energy. Notwithstanding, empirical evidence shows that more developed countries and the more democratic ones tend to diversify the energy sources and increase investment in modern renewable energies (see, e.g., [12,178]). In fact,

the process of development itself leads to a transition toward more investment in renewable energies, which reveals an optimistic view. If that empirical trend continues in the future, policy makers might be led to believe that, with only economic and political development, the energy transition will occur almost automatically. However, both empirical evidence and our review of the literature reveal a lack of concern with the social effects of adoption of renewable energies regarding income distribution, employment, and poverty rates, which is in clear contradiction to the United Nations Sustainable Development Goals. Moreover, the fact that the research relating renewable energies and sustainability in developing countries almost ignores modern renewable energies is also at odds with the implementation of those more modern and diversified renewable energies in the process of development, as empirical evidence has shown. It seems that the scientific community needs to accelerate research on modern renewable energies in developing (and poor) countries and on the social dimension in order to accelerate the transition through the energy ladder and help to fulfill the United Nations Sustainable Development Goals.

All of this underscores that energy plays an important role in the sustainability equation. In fact, many researchers concluded that energy, mainly RE, is fundamental to meet an important part of the world's energy demand and is needed to achieve sustainable development and create business opportunities, along with the wealth and employment that they bring [179]. The relevance of renewables in the nowadays energy mix is clear. In fact, their market share is growing strongly. In 2015, 17.5% of the world's final energy consumption came from those sources, of which 9.6% was produced by modern renewable systems using wind, solar, geothermal, bioenergy, and hydropower. In addition, the share of RE in power generation grew to 22.8% in the same year [180], and there are extensive concerns about preserving the environment and using renewables as sustainable energies. Renewable energy and sustainability will thus remain a popular and strategic topic in the future.

## 6. Conclusions and Research Prospects

The literature bridging renewable energies and sustainability was analyzed, providing the first critical literature review on the issue that considers the three dimensions of sustainability simultaneously. To this end, we reviewed and analyzed 316 articles published from 1997 to 2019.

To guide the analysis, three research questions were devised: "Are there differences between the type of renewable energy focused on in the articles and the level of development of countries in which they are analyzed?"; "Are there differences between the dimension of sustainability focused on in the papers and the country's development level?"; and "Are there differences between the type of renewable energies focused on and the dimension of sustainability analyzed in the articles?"

Most works on renewable energies and sustainability are concerned with environmental problems, mainly those focusing on biomass and wind–solar sources of energies and attempt to find solutions to make the renewable energy production supply chain more sustainable using suitable policies and investments in the economy. In fact, some authors alert the scholarly community to the importance of a set of factors that seems to be responsible for the success of the renewable energy production: (i) the optimization of supply chains and logistics management [181]; (ii) the capacity and location of supply, collection, and processing centers [182]; and (iii) the supplier selection and the SC network design [183]. Regarding the type of renewable energies most focused on in the sample articles, biofuel appears as the most examined followed by biomass, with biodiesel as the least investigated.

The most productive countries in terms of renewable energies and sustainability concerns in the sample are USA, Brazil, European Union, China, and UK. The research community that studies the relationship between renewable energies and sustainability covers 55 countries, most of which are regarded as high development countries. In another work about the economy, energy, and environment [184], also using a bibliometric analysis as methodology, the same countries were identified as the main contributors to the development of that research field: China, USA, UK, and some European Union countries (Italy, Germany, Spain, France, and the Netherlands). The important contribution of the European Union to the advances in renewable energies and sustainability derives

also from the Energy Union Package [185] in 2015. This is a Framework that gives priority to energy policies being supported in the following five priority areas: (1) decarbonization; (2) energy efficiency; (3) internal energy markets; (4) energy security; and (5) research, innovation, and competitiveness. This will lead to the adoption of national policies and monetary incentives motivating the research on renewable energies (energy efficiency) and sustainability (decarbonization). In China, the research topic of renewable energies and sustainability has also increased, supported mainly by the Energy Research Institute of China and the Special Fund arrangement created in 2006, which offers subsidies and grants to renewable energy players such as manufacturers and research institutions in the production and technological innovation of renewable energies with sustainability concerns. Moreover, China's National Mid- and Long-term Plan for Science & Technology (2006–2020) has also been an important driver in this increased tendency [186].

In USA, several energy efficiency and renewable energy programs are responsible for supporting much of the research published in top journals, for example the Renewable Energy Research and Development Program provides financial assistance to conduct balanced research and development efforts in renewable energies and has existed since the 1970s [187].

The main conclusions are as follows. First, the variety of renewable energies explored in the literature that address cases in developed countries is greater than the variety of renewable energies analyzed in the context of developing countries. In the sample articles focusing on low-income countries, biomass and hydroelectric energy are the only two types of renewable energies that have been studied. This resembles the empirical evidence on the energy ladder, according to which more developed countries invest more in more diversified renewable energy sources in their energy mix. This result also reveals the difference between the research unit focused on by academics and reality. According to the 2018 report on Global Trends in Renewable Energy [188], developing countries made up more than 60% of investment in renewable energy of the global investment, while the share of developed economies is only around 37%. This would indicate that most of the cases focused on by academics should be in developing countries, but this was not the case. According Apergis [189], the choice of examining developed countries instead of developing countries is due to the fact that countries with higher average income tend to care more about the environment than those with lower average income, thus the study of renewable energies and sustainability is more appropriately undertaken in these countries.

Second, this study also supports that the environmental dimension of sustainability is the overwhelming concern among studies addressing countries of all development levels. In fact, even for lower income countries, economic sustainability concerns are a minority among all the literature reviewed and social sustainability concerns is not representative. Third, the wind–solar type of energy is the most focused on in the sample articles, with the environmental sustainability alone or with the economic sustainability as the most investigated. This reflects what is observed in reality and reported by the Renewables 2019 Global Status Report [190], which states that the global renewable energy sector is on an upward trend with the wind and solar sources of energies increasing their shares. Moreover, wind and solar energy systems are natural complements since solar power concentrates during the daytime, whereas wind power has greater output at night [191,192]. The main conclusion that can be drawn is that the environmental dimension of sustainability alone or with other sustainability dimensions is the most explored and the social dimension of sustainability is the least. Biodiesel is clearly the exception to the pattern. In the articles focusing on this energy source, social and economic dimensions of sustainability prevail over the environmental dimension.

A further analysis of the results of this research highlights some contradictory views about the issue when coming from the political discourse, empirical evidence, and our own review of the literature. In fact, there are contradictory views coming from the political arena of some of the world's leaders vis-à-vis the Sustainable Development Goals of the United Nations. However, although empirical evidence points to a one-way path that combines development and the energy mix with a high emphasis on modern renewable energies, the literature review reveals a lack of research on

developing countries and those modern sources of renewable energies. Furthermore, it reveals a lack of analysis on the relationship between renewable energies adoption or investments and social dimensions such as poverty, inequality, and employment.

Summing up, the state of the art of the literature may induce a further delay for developing countries to adopt cleaner technologies in their energy production. As these findings are independent of the authors' affiliation, there is a need and an opportunity to re-focus the literature on studying the implementation of cleaner renewable energies in developing countries. Considering the economic and demographic projections, this re-focus of the literature and the potential consequences for increasing the renewable energy production in developing countries is crucial to achieve the Sustainable Development Goals. Energy for sustainable development has been one of the most popular topics in the literature and promises to remain a popular topic in the future also attending to COVID-19. The pandemic has created the biggest global crisis, sending shock waves through health systems, economies, and societies around the world. Faced with an unprecedented situation, governments are focused on bringing the disease under control and reviving their economies. However, the energy sector is also severely affected by this crisis, which has slowed transport, trade, and economic activity across the globe, bringing the generation of energy from fossil fuels to a breaking point. Global energy demand dropped to levels not seen in 70 years and the International Energy Agency (IEA) has estimated that overall energy-related emissions will decrease by 8% for 2020. This represents an important advantage for the environment but also a challenge for the renewable energy sector. It will be interesting and strategic in future works to explore the influence of COVID-19 on the renewable energy sector.

**Author Contributions:** S.G.—Conceived and designed the analysis. T.S.—Performed the analysis. M.S.—Collected the data and performed the analysis. All authors have read and agreed to the published version of the manuscript.

**Funding:** FCT (Fundação para a Ciência e Tecnologia), Project UIDB/05037/2020.

**Conflicts of Interest:** There is no conflict of interests.

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
