# Peer review of "Renewable Energy and Sustainability from the Supply Side: A Critical Review and Analysis"

_applsci, doi:10.3390/app10175755_

Round 1

Reviewer 1 Report

The author could try to add also some information, analysis and perspective on the the potential of tidal power RE;  The authors applied a simple but effective method of analysis of the papers available through appropriate tools and also the correlations leading to the conclusions are reliable.

Author Response

Dear Editor and Reviewer

We would like to thank the Editor and the Reviewers for carefully examining our work and for providing us with the opportunity of revising and improving the manuscript. We have addressed all the comments and suggestions of yours and the Reviewers and modified the paper accordingly. All the modifications are marked with track changes in the revised manuscript in order to facilitate the review process. Thank you very much. The improvements made in the paper are the following:

Answers to Reviewer #1

1 - The author could try to add also some information, analysis and perspective on the potential of tidal power RE; The authors applied a simple but effective method of analysis of the papers available through appropriate tools and also the correlations leading to the conclusions are reliable.

Answer: Thanks a lot for your positive evaluation of our manuscript.

In the introduction section the following was written: “There is another source of RE which is the tidal power that besides contributes to reduce the dependence on fossil fuels it has a negative hydro-environmental impact since the tidal turbines alter ambient flow patterns because of the extraction of Kinetic energy [32]”..

Forward in the Introduction section the following was also written: “Also, [36] argue that besides the tidal and waves energy plants have been considered as green technologies, since they do not alter the climate, conserve resources, have no harmful effect on human health or ecosystems, and are less harmful to the environment than conventional means of energy generations, an assessment of the amount of metal used by these technologies, however, shows an impact respectively 11 and 17 times higher than for coal- and gas-based power generators.”

Yours sincerely,

Susana Garrido, Tiago Sequeira and Marcelo Santos

Reviewer 2 Report

Thank you for reviewing this paper. It is very interesting critical literature review about renevable and sustainability issues. Authors gatheren outstanding number of previous papers, and based on them introduced interesting conclusions.

I suggest gropup research questions in one paragraph, and longer explanations to them move before research questions. In my opinion it will be more readable.

One more suggestion is to extend conclusions with paragraph about Covid-19 influence for future energy demand and renevable energy production. The paper will be very actual with this paragraph.

Good luck with your paper!

Author Response

Dear Editor and Reviewer

We would like to thank the Editor and the Reviewers for carefully examining our work and for providing us with the opportunity of revising and improving the manuscript. We have addressed all the comments and suggestions of yours and the Reviewers and modified the paper accordingly. All the modifications are marked with track changes in the revised manuscript in order to facilitate the review process. Thank you very much. The improvements made in the paper are the following:

Answers to Reviewer #2

Thank you for reviewing this paper. It is very interesting critical literature review about renevable and sustainability issues. Authors gatheren outstanding number of previous papers, and based on them introduced interesting conclusions.

I suggest group research questions in one paragraph, and longer explanations to them move before research questions. In my opinion it will be more readable. Thank you for this suggestion. The research questions were grouped in one paragraphs at the end of the introduction section, just before the definition of the objective of the manuscript. 

One more suggestion is to extend conclusions with paragraph about Covid-19 influence for future energy demand and renewable energy production. The paper will be very actual with this paragraph.

Good luck with your paper!

Answer: Thank you for this suggestion and appreciation. In the last paragraph in the conclusion section, the topic of Covid 19 and the energy sector was introduced.

Yours sincerely,

Susana Garrido, Tiago Sequeira and Marcelo Santos

Reviewer 3 Report

Review report on 886232

Renewable energy and sustainability from the supply side: a critical review and analysis

Authors: Susana Garrido, Tiago Sequeira, Marcelo Santos

Recommendation:  ACCEPT WITH MINOR REVISIONS

INTRODUCTION

The paper presents an extensive review of scientific articles with subject in renewable energy (RE) and sustainability. The Authors' goal is to find if there is a link between the country development level, renewable energy type and the three aspects of sustainability: economic, social and environmental.

Using the scientific literature as investigation field, the orientation of different countries in RE and sustainability domain can be evaluated, since the scientific research is the sparrow tip of economic development.

PAPER CONTENT

The paper is organized in 6 sections, covering 22 pages, 10 figures and 2 tables. The reference section contains 190 titles.

Section 1 - Introduction

In this section the Authors present the basis of the sustainability concept and its three aspects: economic, social and environmental. Are also mentioned scientific researches that proof there is a link between sustainability, renewable energy resources and country development level from where the study came. The Authors focuses on previous state of art studies, emphasis the need to deepen into three directions:

- relationship between the RE types investigated into the articles and developing level of articles' source countries;

- relationship between the sustainability dimension investigated into the articles and developing level of articles' source countries;

- relationship between the RE types and sustainability dimension investigated into the articles and developing level of articles' source countries.

The section ends with a short remainder of paper's structure. In this section 43 references mentioned.

Section 2 - Background

This section contains three subsections:

2.1 Level of countries' development and type of renewable energies.

2.2 The level of countries' development and the sustainability dimensions.

2.3 Relationship between renewable energies and sustainability.

In this section the Authors presents an investigation on how each subject was approached into the scientific articles with focus on RE and sustainability. 21 references were mentioned in this section.

Section 3 - Method and Data

This section contains five subsections:

3.1 Method

3.2 An overview

3.3 Subject Classification of Publications

3.4 Network analysis - keywords co-occurrence

3.5 Most productive countries on renewable energies and sustainability

In this section the Authors present the investigation method used: the bibliometric analysis. Using the VOSviewer software and Scopus database, the articles containing significant keywords for the present work are identified and classified, over 1997-2019 period. Some criteria for selection were used: only the articles from indexed journals that are subject to peer review process; not only English language was considered; the books and conference proceedings were excluded. Other criteria, more specific, were used for inclusion/exclusion of the articles. As result 723 articles were found and after criteria applying 326 were chosen for the present study. Each subsection presents in detail the specific stage. 46 references are mentioned, 6 figures and 1 table are used for results presentation.

Section 4 - The most important topics and analysis

This section contains three subsections:

4.1 Level of countries' development focused on in the papers and the type of renewable energies analyzed

4.2 Relationship between the sustainability dimensions and the level of countries' development

4.3 Relationship between renewable energy and sustainability

In this section the Authors present the results obtained after a deep analyze of the selected papers, looking in the three directions presented in section 1 - Introduction. 98 references are mentioned, 1 table and 3 figures

Section 5 - Further thoughts and critical analysis

In this section the Authors summarize the research results using a graphical representation of the findings on the three directions ("Political and Institutional Discourse", "Empirical Evidence" and "Literature Review") and the implications of these. 11 references are mentioned in this section.

Section 6 - Conclusions and research prospects

The last section of the paper is dedicated to drawing of some conclusion on the presented work, the most important being:

- from the sustainability point of view the main concerning is the environmental problems;

- the developing level of the countries have a great influence on the variety of RE resources explored;

10 references are mentioned in this section.

MERITS

An extensive study on RE and sustainability relationships with the developing level of the countries was performed.

A classification of the most targeted RE resources was presented, with a very clear view of countries which look to develop the RE domain.

The use of bibliometric method and appropriate software in order to identify, classify and analyze the research papers significant for the study goal.

CRITIQUE

Some minor editing errors

OBSERVATIONS ON ERRORS

At page 6, paragraph 2 "is" instead of "id"

At page 10, the last paragraph seems to be unfinished

At page 11, below figure 5, there is an unfinished phrase

At page 13, the table 2 title is on the previous page

At page 14, the caption of figure 7 is on the next page

At page 16, in the downside, there are lines with larger font size

The titles of sections should be written in same way: or with lower case or capitalize each word (section 3 - Method and Data comparing with next section titles)

 CONCLUSION

The paper is in area of interest of the Journal and presents a very interesting study on RE and sustainability.

The method used is accurate and target the most evolved area: scientific research.

As consequence, if the errors will be corrected, the paper can be accepted

Author Response

Dear Editor and Reviewer

We would like to thank the Editor and the Reviewers for carefully examining our work and for providing us with the opportunity of revising and improving the manuscript. We have addressed all the comments and suggestions of yours and the Reviewers and modified the paper accordingly. All the modifications are marked with track changes in the revised manuscript in order to facilitate the review process. Thank you very much. The improvements made in the paper are the following:

Answers to Reviewer #3

MERITS

An extensive study on RE and sustainability relationships with the developing level of the countries was performed.

A classification of the most targeted RE resources was presented, with a very clear view of countries which look to develop the RE domain.

The use of bibliometric method and appropriate software in order to identify, classify and analyze the research papers significant for the study goal.

Answer: Thank you for your appreciation.

CRITIQUE

Some minor editing errors

OBSERVATIONS ON ERRORS

At page 6, paragraph 2 "is" instead of "id".

Answer: Thank you for identify this error it was changed.

At page 10, the last paragraph seems to be unfinished.

Answer: Thank you for this remarks. The sentence: “Spatial distribution of most productive countries producing renewable energy and having sustainability concerns identified in the articles.” is the title of a sub-section, but you’re right, it seems a normal sentence, so we choose to put it in italic to not confusing readers.

At page 11, below figure 5, there is an unfinished phrase.

Answer: Thank you. The same of previous remark.

At page 13, the table 2 title is on the previous page.

Answer: Thanks. The tile is now just before the table 2.  

At page 14, the caption of figure 7 is on the next page.

Answer: Thanks. It was corrected.

At page 16, in the downside, there are lines with larger font size.

Answer: Thanks. It was corrected.

The titles of sections should be written in same way: or with lower case or capitalize each word (section 3 - Method and Data comparing with next section titles).

Answer: Thank you for this remark. The titles were reviewed.

 CONCLUSION

The paper is in area of interest of the Journal and presents a very interesting study on RE and sustainability.

The method used is accurate and target the most evolved area: scientific research.

As consequence, if the errors will be corrected, the paper can be accepted

Answer: Thank you for your appreciation.

Yours sincerely,

Susana Garrido, Tiago Sequeira and Marcelo Santos
